# Development of a Cost-Effective Multiparametric Probe for Continuous Real-Time Monitoring of Aquatic Environments

**DOI:** 10.3390/s25237110

**Published:** 2025-11-21

**Authors:** Samuel Fernandes, Alice Fialho, José Maria Santos, Teresa Ferreira, Ana Filipa Filipe

**Affiliations:** 1Portuguese Environment Agency (APA) P.I., 2610-124 Amadora, Portugal; alice.fialho@apambiente.pt; 2Administração da Região Hidrográfica (ARH) do Alentejo, 7004-514 Évora, Portugal; 3Institute of Earth Sciences (ICT), Évora Pole, Colégio Luís António Verney, 7000-671 Évora, Portugal; 4Forest Research Centre (CEF), Associate Laboratory TERRA, School of Agriculture, University of Lisbon, 1349-017 Lisbon, Portugal; jmsantos@isa.ulisboa.pt (J.M.S.); terferreira@isa.ulisboa.pt (T.F.)

**Keywords:** water quality, embedded systems, water monitoring, environmental monitoring, datalogger, real-time monitoring, uncertainty analysis, ESP32

## Abstract

**Highlights:**

**What are the main findings?**
We present a low-cost (bill of materials < EUR 1000), open-source, solar-powered multiparameter probe (pH, EC, temperature, water level) with GSM/GPRS telemetry and microSD fallback for long-term unattended monitoring.We quantified uncertainty: expanded uncertainty of ±0.4 pH, with ±56.5/±512/±3200 µS/cm at 1413/12,880/80,000 µS/cm, and ±5.2 cm for water level; precision comes from 1000-sample repeats (e.g., pH SD ≈ 0.004).

**What is the implication of the main finding?**
We obtain a high-frequency pH/EC/T/level time series with stated expanded uncertainties (k = 2), enabling defensible thresholds and targeted confirmatory sampling.We develop an open ESP32-based system with cellular telemetry and microSD, supporting scalable deployments; the next step is a short field co-deployment against certified instruments.

**Abstract:**

Continuous, real-time measurements are essential for informed water resource management and the development of strategies for the protection of aquatic ecosystems. Traditional methods of water quality assessment often fail to adequately capture seasonal trends, and the frequency and rapidity of fluctuations. To address this challenge, a standalone, low-cost (<EUR 1000), autonomous multisensor prototype for remote assessment was developed. The design of the system was optimized with a hardware-centric approach to minimize costs, whilst providing reliability and high precision and accuracy. Based on embedded systems and capable of long-range communication through GSM/GPRS, the device operates with minimal human intervention, ensuring timely data availability for analysis and decision-making. The multisensor instrument determines four important water quality parameters: pH, conductivity, temperature, and water level. Calibration and sensitivity analyses were performed; 1000 measurements per sensor indicated distributions consistent with normality for pH, conductivity, and water level. The results demonstrated high performance in pH measurements (mean: 5.65 on the Sørensen scale, R^2^ = 0.9992, expanded uncertainty: ±0.4), conductivity (R^2^ = 0.9999, expanded uncertainties: ±56.52 to ±3200.00 µS/cm for various standards), and water level (R^2^ = 0.9952, expanded uncertainty: ±5.2 cm). Capable of providing continuous, accurate data at low cost, this multiparameter probe has broad applicability in environmental regulation compliance, pollution control, and sustainable ecosystem management.

## 1. Introduction

The exponential increase in intensive agriculture and urbanization during recent decades in many world regions, exacerbated by climate change, has made water issues increasingly urgent, requiring on-site solutions for continuous monitoring [1,2,3,4]. The assessment of water quality parameters in rivers, reservoirs, and lakes, as defined in standardized protocols such as the Water Framework Directive, Clean Water Act [5], and other policy instruments, typically involves the manual collection of water samples in the field at specific intervals during the year. Water samples are then transported in a controlled environment to a laboratory where the analyses are performed [6]. These methodologies require significant costs and may result in potential sample degradation, transcription errors, and temporal delays [7]. Nevertheless, they fail to achieve continuous monitoring while overlooking the assessment of, for example, episodic pollution events [8,9]. Indeed, brief, intense, and unpredictable environmental changes or events occurring on a temporal scale shorter than the sampling frequency of conventional monitoring programs may go undetected, even if frequently occurring [10,11].

Several key water quality parameters hold significant importance for freshwater ecosystems, such as pH [12,13,14], temperature [15,16,17], conductivity [18,19,20], and water level [21,22]. Fluctuations in pH and temperature can significantly impact aquatic biodiversity [23,24], while conductivity variations, which are associated with the level of dissolved ions, may indicate the presence of contaminants or dissolved solids [25,26,27], affecting water suitability for consumption and ecosystem health [28]. Additionally, changes in water level can indicate shifts in hydrological patterns [29,30,31], which can have cascading effects on aquatic habitats and water availability for various uses [32]. In particular, heatwave events are becoming more intense, long, and frequent in some world regions due to climate change, as is the case for southern Europe, which implies significant environmental changes [33,34,35].

Deploying continuous water quality and quantity measurement stations in remote locations with long-range cellular technology enables long-term monitoring, crucial for comprehending ecological and hydrological changes that unfold over extended periods [36,37,38,39,40]. Real-time monitoring capabilities provide instantaneous data acquisition and transmission [41,42,43,44], allowing for rapid decision-making when facing episodic pollution events [45,46]. Multiparameter systems can be potentially integrated with other monitoring systems, such as weather stations or hydrological networks, satellites, aircraft, and drones, providing a holistic understanding of the water environments [47,48,49,50]. Furthermore, adopting a multiparameter system has been found to substantially reduce the logistical demands and overall costs associated with water monitoring operations, making them less time-consuming and more cost-effective [51,52].

Recently, embedded systems have been developed to possess functionalities for data acquisition, processing, logging, and wireless communication [53,54,55,56,57]. These systems, equipped with features like Wi-Fi, Bluetooth, and GSM/GPRS, enable real-time assessments at a low cost [58,59,60,61]. They also allow multiple analog and digital sensors to be integrated through an analog-to-digital converter [62,63,64] or using digital communication protocols, such as SPI, 1-Wire, and I2C, within a single measurement unit [65,66]. The integration of these capabilities into a compact measurement unit allows for efficient and versatile data collection and analysis in diverse applications.

While low-cost instrumentation is often perceived as lacking the robustness necessary for effectively assessing water quality and quantity parameters, its deployment in the field holds the potential for continuous real-time monitoring [67,68], aiming for immediate detection of pollution events, evaluation of critical trends [69,70], and early warning alerts [71,72], and ensuring the successful management of water bodies [73,74]. Simultaneously, low-cost instruments can support installation of more reliable high-cost monitoring systems in strategic locations when necessary [75,76,77]. Although previous studies successfully demonstrate the feasibility of developing such water monitoring systems, they often lack a transparent metrological sensor characterization, for instance, traceable calibration procedures, uncertainty analysis, and validated operating ranges [78,79]. Additionally, previous studies rarely documented measurement-workflow details or implemented strategies to increase data quality acquisition such as improving the signal-to-noise ratio. In addition, many reports omit LOD/LOQ [79,80] and precision and accuracy definitions aligned with analytical-validation practice [81,82], as well as plans for data fusion over time for network-scale use.

This study reports on the development of a multiparametric probe system designed to optimize performance, accuracy, and cost-effectiveness for real-time water quality and quantity monitoring, concurrently measuring pH, water level, conductivity, and temperature. These parameters provide a holistic assessment of the dynamic state of water bodies over time. Regarding the power source of the system, a solar panel coupled with maximum power point tracking and a lithium battery will ensure self-sufficiency in energy provision. The main goals of the development project were as follows: (i) develop an open-source, <EUR 1000 multiparameter system in the laboratory to understand its performance and validate its capability to measure pH, EC, temperature, and water level; (ii) design a custom printed circuit board (PCB) capable of housing all electronic components of the system, including a GSM/GPRS telemetry module, an on-board microSD to provide data logging as a backup to telemetry, and a solar-powered energy subsystem (maximum power point tracking and a 4000 mAh Li-ion battery), enabling autonomous field operation; (iii) provide a metrological characterization comprising four-point pH calibration with same-cycle temperature compensation (Nernst), three-point EC calibration (cell-constant model), and five-point water level calibration (hydrostatic model), with explicit precision (SD; n = 1000), limit of detection and limit of quantification, and accuracy as expanded uncertainty (coverage factor k = 2); (iv) deliver a documented measurement workflow improving the data quality acquisition that acquires temperature first and implements 10,000 samples for the water level probe to improve the signal–noise ratio. The system’s acquired data will be sent to an online database using long-range communication technology and saved offline on a memory card or other storage and backup system. Stored data can be accessed using a geographical dashboard, equipping decision-makers with the necessary tools to oversee water resources effectively.

## 2. Materials and Methods

### 2.1. Sensor Description

The cost-effective multiparameter system is a compact, lightweight, transportable instrument that is easy to deploy in the field and capable of recording high-quality data and reporting it in real time. The developed system was designed to comprise five major modules (Figure 1). The first module refers to the power source, which integrates a solar panel, a battery, and a power management board. This component can deliver electrical power to sensors and the microcontroller, ensuring that the equipment remains energetically self-sufficient. The second module corresponds to the temperature, pH, water level, and conductivity sensors. The third module refers to processing communication and logging the recorded information. It comprises an embedded system ESP32 TCALL—LILYGO^®^, Longgang Shenzhen, Guangdong, China integrated with a SIM800L for long-range communication, and a microSD card. It also includes a printed circuit board where all the electronic elements are placed. The fourth module consists of a computer server MySQL database, which processes and stores the received data from the multiparameter system. The final module, a Leaflet dashboard, allows for the collected data to be dynamically presented, offering a comprehensive and user-friendly visualization. This approach allows for the rapid detection of water parameter variations in aquatic environments with very high spatial precision and autonomy.

The instrument combines four sensors within a single package (i.e., second module) for water quality and quantity determination, with the possibility of recording data in loco and reporting it in real time. The main advantage of recording the data in loco is that if the long-range communication system fails, the data is not lost since it remains available in the microSD card. The instrument is designed to measure pH, temperature, conductivity (quality parameters), and water level (quantity parameter). All hardware design files, firmware, server code, and analysis scripts are openly available at AQUADAPT_Multiparametric_Probe (GitHub): https://github.com/samuel-q-fernandes/AQUADAPT_Multiparametric_Probe (accessed on 4 November 2025).

#### 2.1.1. pH Sensor

In the proposed multiparameter probe system, the pH measurements are carried out using a Mini Lab Grade pH Probe -ENV-20-pH from Atlas ScientificTM, Long Island City, New York, USA. This sensor can measure from 0 to 14 pH on the Sørensen scale, with a resolution of pH = 0.001 and an accuracy of pH = 0.002. The sensor response time within a probability of 95% is 1 sec, with a working temperature between −5 and 99 °C and a maximum depth of 78 m. The sensor is connected to the ESP32 through the Gravity pH meter V2.1 board, with an accuracy of 0.2 within a range between 0.1 and 1.4, a resolution of 0.1, and a continuous response time. The Gravity™ Analog pH Sensor/Meter-GRV-pH, Long Island City, New York, USA and Electrically Isolated EZO™ Carrier Board, Long Island City, New York, USA are electrically insulated. This allows the microcontroller to record the analog signal from the sensor while mitigating interference and ensuring electrical insulation between components. The recommended calibration time is approximately one year, and the expected lifespan is two years.

#### 2.1.2. Conductivity Sensor

The conductivity determination is achieved using the Mini Conductivity Probe K 1.0-ENV-20-EC-K1.0, from Atlas ScientificTM. The sensor presents a range of measurement between 5 and 200,000 μS/cm, with an accuracy of 2%. The sensor response time within a probability of 90% is 1 sec, with a working temperature between 1 and 110 °C, and a maximum depth of 352 m. The sensor is connected to the ESP32 through an EZO™ Conductivity Circuit, Long Island City, New York, USA with an accuracy of 2%, a range between 0.07 and 500,000+ μS/cm, and a response time of 600 msec. The recommended calibration interval is approximately ten years with an anticipated lifespan of the same duration.

#### 2.1.3. Temperature Sensor

The temperature determination is carried out using a DS18B20 sensor from Dallas Semiconductor, Dallas, Texas, USA, which several authors have utilized and described [83,84,85,86,87,88]. The sensor can measure a range between −55 and +125 °C with an accuracy of 0.5 °C from −10 to +85 °C. It can provide 12-bit temperature readings. The communication with the microcontroller is performed using a 1-Wire digital protocol. A 4.7 kΩ resistor was introduced as a pull-up resistor between the microcontroller Vcc and the temperature sensor data signal pin, ensuring communication stability and power-over-data mode.

#### 2.1.4. Water Level Sensor

The chosen water level sensor was the Gravity Industrial Stainless Steel Submersible Pressure Level Sensor from DFROBOT^®^, Shanghai, China. The sensor presents a measurement range between 0 and 5 m with an accuracy of 0.5%. The MT3608, Shaanxi, China, conversion DC-to-DC step-up was introduced in order to provide a stable voltage between 12 and 36 V for this sensor. The step-up increases the voltage level from the incoming 5 V from the direct USB connection to the minimum requirement of 12 V input of the water level sensor.

#### 2.1.5. Maintenance Requirements and Field Durability

Field deployments expose the submerged sensors to corrosion (oxidation), abrasion, scaling, electrical noise, and biofouling, all of which can degrade the measurement quality and increase operational costs. To prevent the occurrence of biofouling, the probes must be fitted with a copper-based antifouling guard (or copper tape where appropriate) that will help to suppress biofilm growth around sensors and extend their lifespan. A summary of the routine maintenance and verification/calibration frequency for EC, pH, temperature, and pressure sensors used in this study is presented in Table 1.

### 2.2. Printed Circuit Board

A printed circuit board (PCB) with 100 × 100 mm dimensions (Figure 2a) was developed to combine all electronic components that comprise the multiparameter system except for the sensors that are underwater. The PCB supports the development of a compact device with improved signal integrity, reliability, and ease of field deployment. It includes nine pads for placement (Figure 2b), the embedded system, the Gravity™ Analog pH Sensor/Meter-GRV-pH (Figure 3, device 1), the Electrically Isolated EZO™ Carrier Board (Figure 3, device 2), the step-up DC-to-DC converter (Figure 3, device 5), and connectors for temperature, the water level analog interface, and the water level power source (Figure 3, white connectors labeled on the PCB).

### 2.3. Embedded System and Data Storage

The embedded system used in this study features the LILYGO^®^ TTGO T-Call V1.4, which combines a SIM800L module for GSM and GPRS communications with the ESP32 microcontroller (Figure 3, 3). The ESP32 microcontroller is equipped with two energy-efficient Xtensa^®^ 32-bit LX6 microprocessors capable of Wi-Fi and Bluetooth communication. A microSD module and a microSD card capable of storing 16 GB of data are used for logging the measured variables.

### 2.4. Measurement System Workflow

The embedded system presented in this work has been programmed using the Arduino platform version 2.3.0. The developed software tool comprises six main tasks, presented in Figure 4, running in sequential order on the Xtensa^®^ 32-bit LX6 microprocessor. The entire process operates in complete autonomy, eliminating the need for manual intervention at any stage of the process.

In the first task, System Configuration and Initialization, the software and hardware are initialized, which include the communication interfaces to establish communication with the GSM/GPRS carrier and sensors for data collection. The process also includes acquiring date and time, defining the multiparameter system ID, and setting the API key to ensure proper connectivity with the MySQL server.

After initialization, the system transitions into the Data Acquisition from Environmental Sensors phase to determine and record the environmental variables. Water level is read via the analog-to-digital converter using oversampling: 10,000 raw samples are collected per measurement and averaged to improve the signal-to-noise ratio and reduce variability, ensuring higher accuracy and stability.

The temperature is acquired from the DS18B20 sensor interfaced using the 1-Wire communication protocol as it is an integrated 12-bit ADC. Following pH determination, the system records the voltage from the Mini Conductivity Probe K 1.0-ENV-20-EC-K1.0 and the Gravity™ connected to the Analog pH Sensor/Meter-GRV-pH and Electrically Isolated EZO™ Carrier. The analog signal provided by the carrier board is then digitized using the ADC of the ESP32 and later undergoes compensation for temperature fluctuations in postprocessing. Finally, the conductivity is determined using the Mini Conductivity Probe K 1.0-ENV-20-EC-K1.0 connected to the EZO™ Conductivity Circuit. This carrier board provides a digital value converted using a proprietary ADC and interfaced with the ESP32 using the I^2^C protocol.

The third task performed by the system is Data Processing. It involves averaging the 10,000 measurements from the water level sensor; the acquired value in counts is then converted into a voltage $V_{d}$ value using Equation (1) [89]:(1)$$V_{d}=\left(\frac{a_{r}}{4095}\right)\times V_{ref}$$
where 4095 is the maximum resolution of the 12-bit ESP32 ADC, $a_{r}$ is the measured value in counts, and $V_{ref}$ is the reference voltage of the system. The voltage is further used to calculate the current I that follows through the sense resistor (120 ohms) using Equation (2) [89]:(2)$$I=\left(\frac{V_{d}}{120}\right)\times V_{ref}$$

The determination of the depth is then performed using the following relation [89]:(3)$$d=\left(\frac{I-I_{I}}{16}\right)\times \frac{R}{W_{d}}$$
where $I_{I}$ denotes the initial current at 0 mm depth, R is the total depth measurement (mm), which, for this sensor, is 5000 mm, 16 is introduced for scaling, with this value provided by the manufacturer, and $W_{d}$ is the water density.

To determine the pH, the raw value in counts acquired from the sensor is converted into voltage $V_{pH}$ using Equation (4) [90]:(4)$$V_{pH}=\left(\frac{a_{pH}}{4095}\right)\times V_{ref}+offset$$
where $a_{pH}$ is the analog value from the ADC in counts and an offset is added to compensate for the non-linearities of the ESP32 ADC. The final value is used to determine the final pH value by using the methodology of pH and temperature compensation outlined in Section 2.7.1 along with the calibration sensor.

For the conductivity and temperature sensors, the measured value is acquired directly from the digital sensor.

In the fourth task, Real-Time Data Transmission, the data is transmitted to a remote server through the communication module. The multiparameter system establishes a cellular connection GSM/GPRS using the SIM800L module integrated into the embedded system (LILYGO^®^), ensuring a long-range communication capability. The API key, multiparameter probe ID, date, time, sensor data, and observations are packed into an HTTP post request and transmitted to a remote server, enabling real-time data logging and monitoring.

The fifth task, Local Data Logging for Redundancy, was added to the system to prevent loss of data in the event of a network failure or interruption of data transition to the server. In this process, the multiparameter probe ID, date, time, sensor data, and observations are stored in the microSD card in csv format. The data recorded can be retrieved by connecting the microSD card to a personal computer.

In the final task, Energy-Efficient Power Management and Sleep Mode, the system enters a deep-sleep mode once the data is collected. Before sleep, the SIM800L is also powered down. These two actions ensure that the system minimizes the energy consumption, allowing it to operate over longer intervals of time, considering that it is powered using a battery and a solar panel. After the 10-min deep-sleep period, the multiparameter probe exits low-power mode and reinitializes all the procedures, operating in a continuous loop. Figure 5 presents the power consumption of the system for a period of approximately 70 min of operation.

### 2.5. Power Source and Storage

The powering unit includes a photovoltaic solar panel with a maximum power of 5 W, maximum voltage of 6 V, and current of 833 mA. The panel is connected to a power manager module featuring maximum power point tracking. This module can provide up to 900 mA charging current and 3.7 V output. The energy is stored using a Lithium-Ion polymer battery with a maximum output voltage of 3.7 V and a maximum current of 4000 mAh.

### 2.6. Sensitivity Analysis for Combined Uncertainty Evaluation

A sensitivity analysis was performed to assess the combined uncertainty of the pH, water level, and conductivity sensors. The methodology combines the uncertainty specified by the manufacturer and the uncertainty determined from the experimental measurements using the following relationship [91]:(5)$$u_{c}(y)=\sqrt{\sum_{i=1}^{N}{\left(c_{i}\cdot u_{m}(x_{i})\right)}^{2}}+\sum_{j=1}^{M}{\left(c_{j}\cdot u_{p}(x_{j})\right)}^{2}$$
where $u_{c}(y)$ denotes the combined standard uncertainty of the output quantity y, $c_{i}=\frac{\partial f}{\partial x_{i}}$ is the sensitivity coefficient of the input $x_{i}$, determined as the partial derivative of the functional relationship f(x), which, for direct measurements, is assumed to be 1, $u_{m}(x_{i})$ denotes the manufacturer-specified standard uncertainty, and $u_{p}(x_{j})$ is the experimental measurement standard uncertainty. Expanded uncertainties *U* are reported at ~95% coverage using a coverage factor k = 2 (per GUM).

This is expressed as(6)$$U=k{\cdot u}_{c}(y)$$

### 2.7. Calibration Methods for pH (With Temperature Compensation), Electrical Conductivity, and Water Level

#### 2.7.1. pH Temperature Compensation and Calibration

The carrier board Gravity™ Analog pH Sensor/Meter-GRV-pH does not implement a temperature compensation for the determination of pH, which helps reduce manufacturing costs. The method uses the Nernst equation (Equation (7)) [92,93,94] to correct the pH value in accordance with the measured temperature from the DS18B20 sensor.

For each measurement cycle, we read temperature T (K) from the sensor in T (°C):(7)$$T=T_{c}+273.15$$
and compute the instantaneous Nernst slope, according to Equation (8).(8)$$S\left(T\right)=1000\times \frac{2.303RT}{F}\;(mV\;per\;pH)$$
with *R* denoting the gas constant (8.314 J/(mol·K)), and *F* the Faraday constant (96,485 C/mol). We then model the reference cell; the pH sensor uses the silver chloride electrode with a standard electrode potential E^0^ = 0.222 V [95]:(9)$$E=E^{0}\left(T\right)-S(T)\times pH$$
so that(10)$$pH=\frac{E^{0}\left(T\right)-E}{S(T)}$$
where *E* denotes the measured cell potential of the pH electrode, *n* is the number of electrons transferred in the redox reaction (for a pH electrode, typically 1), *[H^+^]* is the concentration of hydrogen ions (protons) in the solution, and *[H^+^]_ref_* represents the reference concentration of hydrogen ions (typically 1 mol/L).

The calibration process was conducted to establish a reliable correlation between the sensor output voltage coming from the sensor and the actual pH values of the standard solutions. The pH sensor was calibrated using a set of pre-manufactured buffers. The pH sensor underwent calibration at four specific reference points: pH 4, 7, 9, and 10 [96]. These reference points were chosen to cover a spectrum of acidity and alkalinity levels found in aquatic environments. From the Nernst equation, the cell potential for the four pH standards has been derived: 0.454 V, 0.629 V, 0.745 V, and 0.803 V, respectively. Standard solutions from Certipur^®^, Darmstadt, Germany, Buffer were employed for calibration, with compositions traceable to Standard Reference Material from NIST and PTB, characterized by an uncertainty of ±0.002 at a temperature of 20 °C.

#### 2.7.2. Conductivity Sensor Calibration

In contrast to the pH sensor, the carrier board of the conductivity sensor does not allow direct access to the raw measured values. The temperature compensation of the conductivity is also determined in the carrier board. Therefore, in this study, a calibration procedure was conducted for the conductivity sensor in accordance with the guidelines of the manufacturer. The recommended standard solutions are at 12,880 µS/cm and 80,000 µS/cm, both sourced from Atlas Scientific. These solutions present an uncertainty of ±5 µS/cm at 25 °C and are NIST-traceable. In addition, a standard solution at reference point 1413 µS/cm was introduced. This standard was sourced from ASTM, West Conshohocken, United Sates of America (USA), with the reference D1125 and presents an accuracy of 1% at 25 °C. Introducing a lower conductivity standard is intended to reduce measurement inaccuracies for water bodies, which are expected to exhibit low conductivity levels.

#### 2.7.3. Water Level Sensor Calibration

The performance of the water level sensor was determined in a small water storage reservoir with 110 cm depth. The sensor was introduced at various levels: 30 cm, 40 cm, 75 cm, and 100 cm. A SEBA electric contact meter type KKL was used as a working standard to determine the proper depth and compare the measured values between both instruments. The instruments were lowered into the water at depths up to 100 cm.

### 2.8. Measurement Distribution Analysis

A sensitivity analysis was conducted to characterize the Probability Density Function (PDF) for pH, conductivity, and water level sensors to determine the measurement uncertainty and the likelihood of specific values, and increase the reliability of measurements. This analysis enabled us to quantify the inherent variability of these sensors, using the standard deviation as an indicator of accuracy and precision, and to detect any potential bias in the measurements.

### 2.9. System Integration and Housing

The final design of the multisensor system is illustrated in Figure 6. This setup integrates a waterproof enclosure to safeguard the sensitive electronic components. By providing a robust barrier against moisture, dust, and other environmental contaminants, the enclosure ensures the longevity and reliable performance of the system. Including a 7-m cable in the multisensor system enables the sensors to be positioned in distant locations. This feature is particularly important during the rainy season, when water levels may rise unexpectedly. The 6V solar panel provides the energy to ensure the autonomous operation of the system, combined with the MPPT and the battery. The results indicate that the energy storage system can sustain two full days of remote transmission at 10-min intervals without requiring solar power recharging.

### 2.10. Dashboard of the Multiparametric Probe System

A Leaflet-based platform version 1.6.0 was used to develop a geographic dashboard that offers an interactive interface for viewing and downloading the data stored in a MySQL database (Figure 7). With the ability to monitor and analyze key water indicators such as pH, conductivity, temperature, and water levels, this dashboard serves as a central tool to save all sensor data. It also allows users to detect individual episodes, seasonal oscillations, and long-term patterns. A collection of tools necessary for making well-informed decisions is also given to users and decision-makers via the dashboard.

## 3. Results and Discussion

### 3.1. Probability Density Function of Repeated Measurements

The PDF assessment of the Mini Lab Grade pH Probe -ENV-20-pH sensor was conducted by running 1000 measurements in a water sample at room temperature, and results are presented on the Sorensen pH scale (Figure 8a). A Jarque–Bera statistical test allowed the verification of a normal distribution for the sensor with a *p*-value of 0.05. The average pH measured by the sensor was 5.657, with a standard deviation of 0.004.

Similarly, the Mini Conductivity Probe K 1.0 (ENV-20-EC-K1.0) was tested with 1000 measurements of a water sample at room temperature (Figure 8b). According to the Jarque–Bera test, the readings are consistent with normality (*p* = 0.15). The average conductivity measured was 242.62 µS/cm with a standard deviation of 0.14 µS/cm, demonstrating reliable performance with low variability. The Jarque–Bera statistical test was also conducted on 1000 measurements collected from the water level sensor (Figure 8c), indicating that the sensor follows a normal distribution for a *p*-value of 0.30. The average water level measured was 40.62 cm with a standard deviation of 0.37 cm. The sensitivity analysis confirms that all sensors exhibit normally distributed measurement data, critical for ensuring reliability in practical applications. The low standard deviations across all sensors indicate high precision, making them suitable for accurately monitoring pH, conductivity, and water levels.

### 3.2. Calibration Results and Uncertainty for pH (Temperature-Compensated) and Water Level

The relation between the measured voltage and the standard solution at different pH values was found to vary linearly at room temperature. The observed relation shows that as pH levels increase, voltage consistently decreases (Figure 9a). The instrument exhibited a high coefficient of determination (R^2^ = 0.9992, F-statistic = 2874, and *p*-value = 0.000347), indicating a strong correlation between voltage and pH. The expanded measured uncertainty, considering sensor accuracy, carrier board accuracy, and the standard deviation from distribution analysis, was ±74.7 mV, corresponding to a pH variation of ±0.4. The limit of detection (LOD) and limit of quantification (LOQ) for pH were determined to be approximately 0.012 and 0.040 pH, respectively.

The water level sensor could determine fluctuations in the water column as demonstrated by the high linear correlation (R^2^ = 0.9952, F-statistic = 630, and *p*-value = 0.00013) with the water level measurement tape (Figure 9b). The coefficient of determination showed a very high concordance for the measurements performed using both instruments in the same environment. The performance evaluation incorporated the manufacturer-specified accuracy and empirical data from the testing environment. The sensor’s expanded uncertainty was calculated by integrating the accuracy provided by the manufacturer’s specification (0.5%) with the standard deviation derived from the distribution analysis, which resulted in an uncertainty of 5.2 cm. The LOD and LOQ for water level were determined to be approximately 1.11 cm and 3.70 cm, respectively.

### 3.3. Conductivity Calibration Results and Uncertainty

A correlation was observed between the conductivity standard solutions and the data recorded by the sensor (Figure 10a). This strongly suggests a consistent linear trend between the two variables, highlighting their interdependence and the ability of the sensor to accurately measure conductivity, presenting a coefficient of determination (R^2^ = 0.9999, F-statistic = 16,623,803, and *p*-value = 0.00015) showing a very high correlation. The expanded uncertainties by integrating the 2% accuracy provided by the manufacturer and the determined standard deviation from the distribution analysis for the measured values 1413, 12,880, and 80,000 µS/cm were 56.52, 512.20, and 3200.00 µS/cm, respectively. A detailed zoomed-in view of the calibration points 1413 and 12,880 µS/cm is presented in order to better understand the uncertainty (Figure 10b and Figure 10c, respectively). The LOD and LOQ for the electrical conductivity probe were determined to be approximately 0.42 µS/cm and 1.40 µS/cm, respectively.

A summary of the key characteristics of the developed instrument (AQUA V1) and other water quality monitoring systems reported in the literature as well as commercially available systems is described in Table 2. Overall, the studies implementing low-cost solutions for water monitoring often fail to present an uncertainty analysis, leading to limitations in reliability and applicability of their findings. The cost analysis highlights the cost-effectiveness of the AQUA V1 compared to other systems. Despite the lower cost of the [67] system, it lacks precision in pH measurement compared to AQUA V1. Additionally, AQUA V1 includes advanced functionalities like GSM/GPRS connectivity, which proves to be advantageous in areas with limited LoRaWAN network infrastructure and eliminates the cost associated with the deployment and maintenance of custom gateways. Additionally, most of the presented instruments lack a graphical user interface for smartphones or web-based dashboards, restricting the users that rely on mobile and online platforms from real-time data visualization and decision-making.

## 4. Conclusions

This research aimed to develop and calibrate a low-cost, multisensor probe system prototype for continuously monitoring water quality and quantity parameters in water bodies. Our prototype follows previous recommendations to improve monitoring systems and assessments in the context of the Water Framework Directive, as traditional methods relying on manual sampling and laboratory analysis face challenges in providing real-time data and thus fail to enable the rapid assessment of episodic pollution events such as stormwater, runoff, and industrial discharge.

The results show that the newly designed multiparametric probe is quite effective in assessing the quality and quantity of water with good precision for key parameters: pH, temperature, conductivity, and water level. Across 1000 measurements for each sensor, Jarque–Bera showed normality for pH, conductivity, and water level. The performance of the used sensors in the prototype indicated a strong correlation between the recorded values and standard solutions, presenting high coefficients of determination (R^2^ > 0.99) and a high reliability in the measurements. Specifically, a linear relation between voltage and pH levels was observed for pH measurements with minimal uncertainty. The water level sensor was able to accurately detect fluctuations in the water column, showing a strong agreement (R^2^ > 0.99) between the measured values and the established working standards. Similarly, the conductivity sensor calibration procedure showed a consistent linear trend and high correlation coefficient (R^2^ > 0.99) between the conductivity of standard solutions and recorded data.

Continuous developments, such as the integration of new sensors capable of measuring dissolved oxygen, turbidity, and nitrate levels, along with further system miniaturization, further improve prototype efficiency and applicability in diverse aquatic environments. Additionally, a data fusion model capable of integrating real-time data with historical information based on a state-space/Kalman filtering pipeline in which each sensor reading is a noisy observation of a latent water quality state should be developed. This will enable ecosystem health assessments to be made remotely in the context of previous events, which is critical in supporting quick and more informed decision-making for water resource management, pollution control, and biodiversity conservation. The system has undergone a series of extensive laboratory tests. While the probe enables low-cost, multiparameter monitoring, it requires periodic maintenance and recalibration, its stated uncertainty applies only within the calibrated range, and field co-validation is still needed to confirm in situ bias and drift; telemetry and power constraints may limit unattended operation. Future work will include short periods of field deployment to quantify in situ performance, stability, bias, and drift, assess fouling, and confirm power autonomy under real conditions. These trials will also validate the system under real-world conditions, enabling assessment of performance of the sensors under realistic variations in temperature, water flow rates, and contamination loads. Once fully validated, datasets can support routine monitoring, trigger targeted compliance and enforcement when thresholds or anomalies occur, and inform risk-based design, baselining trend assessments, and hence evidence-based decisions. A subsequent PCB revision will add a MOSFET to fully isolate the sensor rails during deep sleep, reducing energy consumption. Furthermore, the developed device demonstrates broad applicability across various industries including oil, gas, mining, textiles, chemicals, and the food processing industry, helping in managing contaminants and complying with discharge regulations. When used in water treatment plants, the system can optimize real-time chemical dosing and ensure that treated water meets safety standards for the monitored pollutants. This multisensor system can also be introduced on a wireless sensor network (WSN), a network of spatially distributed sensors for environmental monitoring (water quality and quantity), thus increasing the coverage of monitoring efforts across large and remote areas.

## Figures and Tables

**Figure 1 sensors-25-07110-f001:**
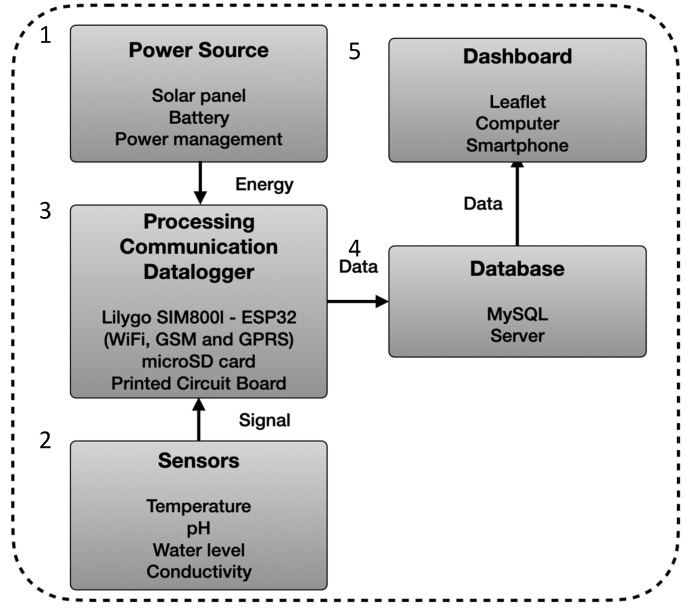
System architecture of multiparameter probe, showing sensors, embedded controller (ESP32), GSM/GPRS telemetry, power subsystem, microSD logging, and database/dashboard.

**Figure 2 sensors-25-07110-f002:**
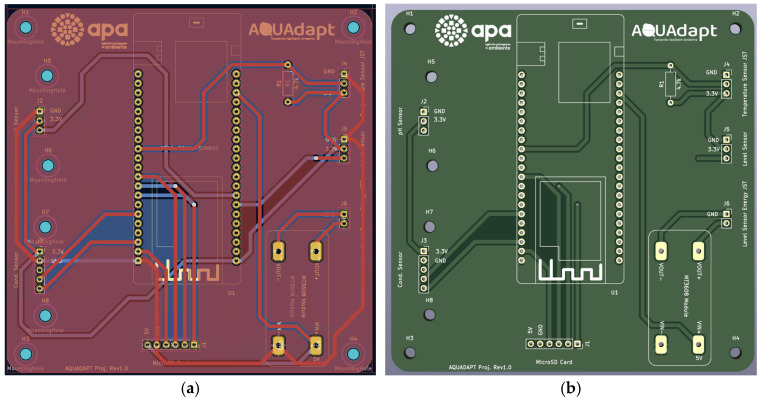
Overview of the printed circuit board (PCB): (**a**) circuit schematic; (**b**) 3D rendering of the assembled PCB.

**Figure 3 sensors-25-07110-f003:**
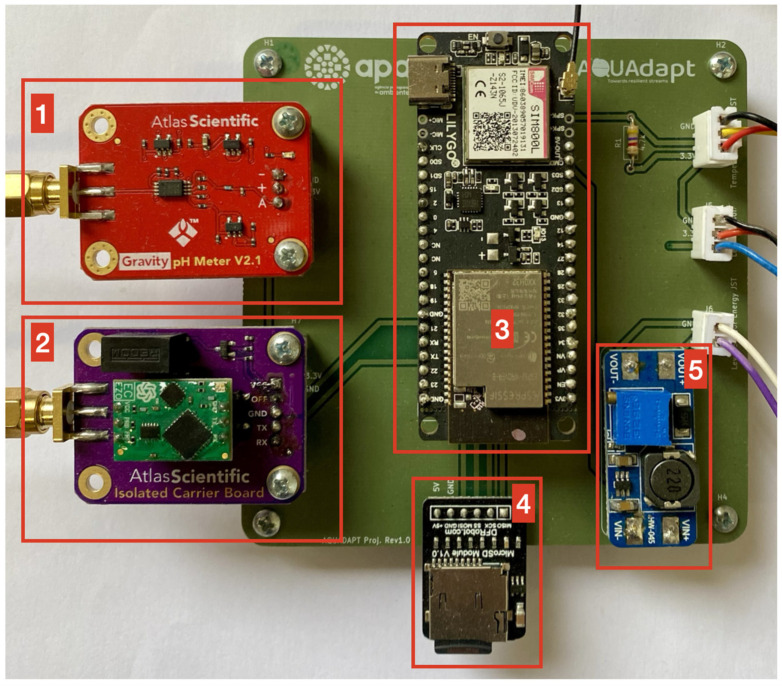
Front view of multiparameter probe with PCB and electronics: (1) Gravity™ pH front-end (GRV-pH carrier); (2) Atlas Scientific EZO™ isolated carrier; (3) LILYGO^®^ TTGO T-Call v1.4 controller; (4) microSD logging module; (5) DC–DC step-up converter.

**Figure 4 sensors-25-07110-f004:**
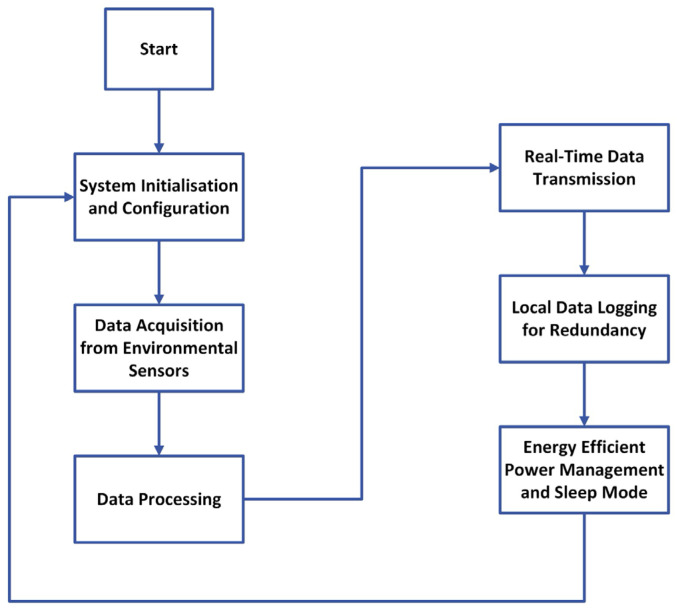
Sequential workflow of the environmental multiparameter real-time monitoring system operating in a loop.

**Figure 5 sensors-25-07110-f005:**
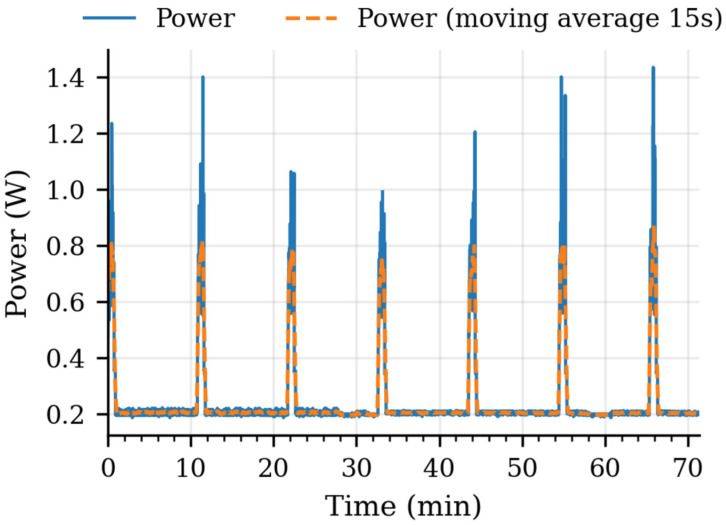
Power profile of the multiparameter system during ~70 min. Short bursts correspond to measurement/communication, followed by deep-sleep intervals after the SIM800L shutdown. A 15 s rolling mean is shown for clarity.

**Figure 6 sensors-25-07110-f006:**
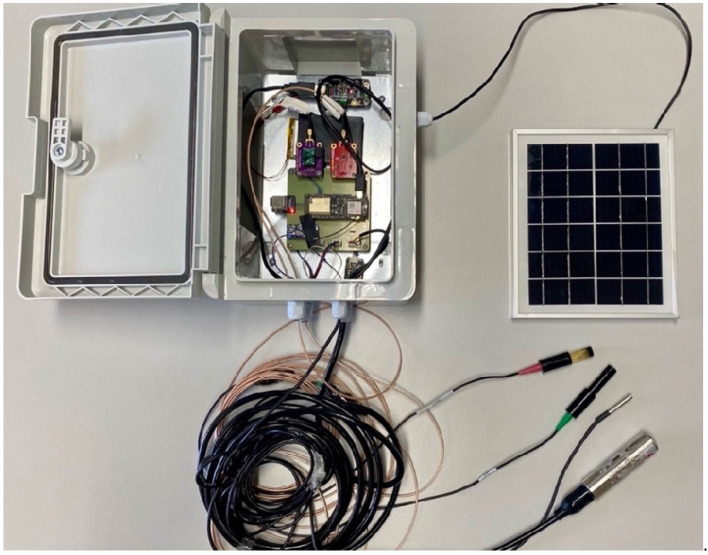
Hardware overview of the multiparameter monitoring system, showing the enclosure, sensor bundle, PCB, and 6 V photovoltaic panel.

**Figure 7 sensors-25-07110-f007:**
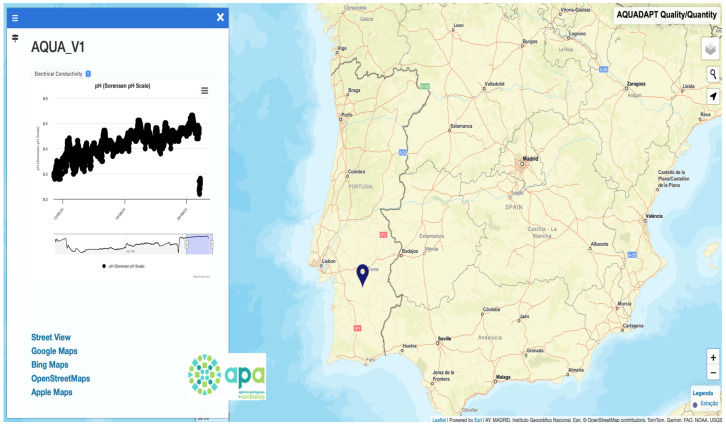
Web dashboard of the multiparameter sensor system showing synthetic pH measurements at site AQUA v1 (Alentejo, Portugal) from 11 to 27 August 2024.

**Figure 8 sensors-25-07110-f008:**
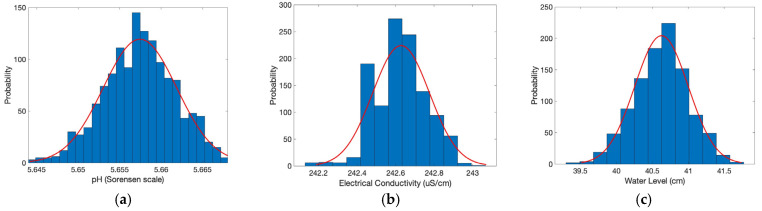
Probability Density Function (PDF) determination of (**a**) pH sensor, (**b**) conductivity sensor, and (**c**) water level sensor. Normal curve distribution: blue bars; best-fit distribution line: red.

**Figure 9 sensors-25-07110-f009:**
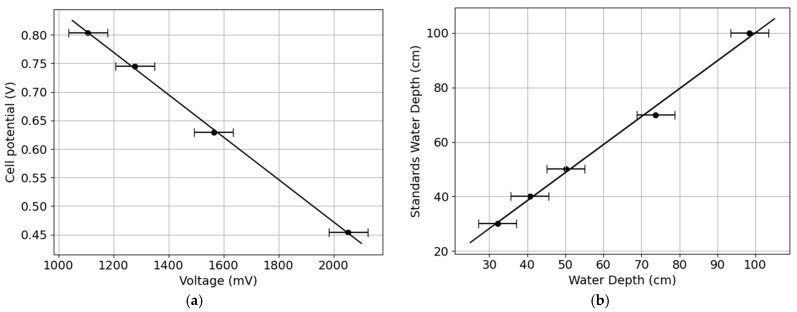
Calibration curves: (**a**) measured electrode voltage versus pH standards (linear fit: slope −0.0004 V/pH, intercept 1.2155 V; R^2^ = 0.9992); (**b**) sensor-measured water depth versus reference depth (linear fit: slope 0.0064, intercept 17.0892 cm; R^2^ = 0.9952).

**Figure 10 sensors-25-07110-f010:**
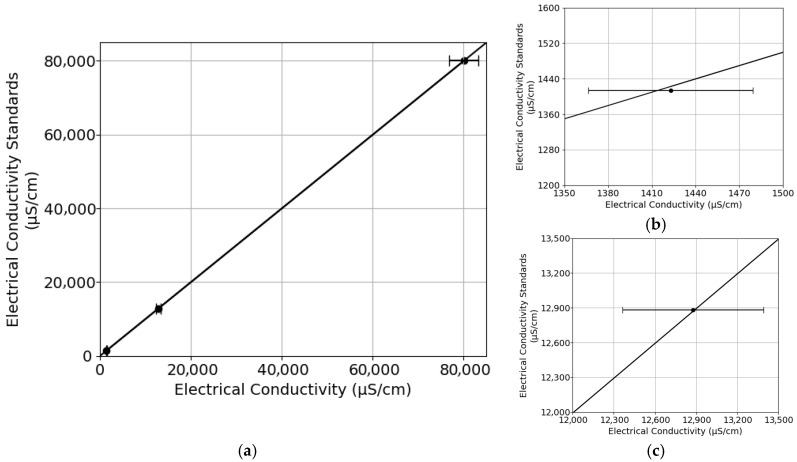
Calibration of the conductivity sensor: (**a**) overall response with expanded uncertainties k = 2, ≈95% coverage; (**b**) zoomed-in view at 1413 µS/cm; (**c**) zoomed-in view at 12,880 µS/cm. Linear regression: slope = 1.0266, intercept = −2.5964 µS/cm (R^2^ = 0.9999).

**Table 1 sensors-25-07110-t001:** Maintenance and durability summary for field deployments.

Sensor	Cleaning Frequency	Verification/Calibration	Field Durability (Corrosion/Physical)
E.C. Conductivity	Rinse after retrieval. Quarterly chemical clean (EC-safe cleaner); every 2–4 weeks in high fouling.	Annual verification against standards; recalibrate only if verification fails after cleaning. Always apply temperature compensation.	Electrodes robust; housings/cables are usual failure points. Avoid galvanic pairs; isolate from copper. Protect cable with strain relief.
pH	Rinse after use; store wet. Quarterly chemical clean (or on drift); monthly in high-fouling/harsh chemistry.	Monthly verification; annual calibration in benign media, up to monthly in harsh acids/bases. Temperature compensation recommended.	Glass bulb/junction fragile; protect from impact/abrasion and thermal shock.
Temperature	Minimal; if submerged, quarterly wipe to remove films; monthly if heavy fouling.	Annual verification (ice bath or calibrated meter); recalibration rarely required; prefer 3-/4-wire.	Element is robust; cable/connector ingress and flex fatigue are main risks. Good corrosion resistance.
Water Level	At six months, gentle rinse of diaphragm with low-pressure water/air; every 2–4 weeks in high-fouling waters. Never scrape piezoelectric sensor.	Semiannual verification (static head or reference gauge); increase to quarterly in harsh service; zero-check after cleaning.	Use IP68 for continuous submersion. Protect piezoelectric sensor from impact/abrasion with a cage.

**Table 2 sensors-25-07110-t002:** Feature and performance comparison: AQUA V1 and selected water quality monitoring systems.

Water Quality Measurement System	AQUA V1	[97]	[67]	[98]	HACH SC1000 Multi-Parameter	HANNA Multiparameter HI98194	Aqua TROLL 800
Cost	<USD 1000	<GBP 5000	USD 240		USD 10,000 to USD 15,000	USD 2003 to USD 2645	USD 7000 to USD 8000
Measurement Principle: pH	Potentiometry	Potentiometry	Potentiometry	Potentiometry	Potentiometry by differential electrode	Potentiometry	Potentiometry
Measurement Principle: Conductivity	Electrical conductance	Electrical conductance		Electrical conductance	Inductive (Toroidal)	Electrical conductance	Electrical conductance
Measurement Principle: Water Level	Piezoresistive				Pressure transducer with stainless steel diaphragm		Piezoresistive; Ceramic
Response Time: pH	95% in 1 s		<1 min				T63 < 3 s, T90 < 15 s, 95 < 30 s
Response Time: Conductivity	90% in 1 s						T63 < 1 s, T90 < 3 s, T95 < 5 s
Response Time: Water Level							T63 < 1 s, T90 < 1 s, T95 < 1 s
Range: pH	0–14 pH	0–14 pH	0–14 pH	0–14 pH	−2.0 to 14.0 pH	0.00 to 14.00 pH	0 to 14 pH units
Range; Conductivity	5 to 200,000 μS/cm	1000 to 55,000 μS cm^−1^		0 to 20,000 μS/cm	200 to 2,000,000 μS/cm	0 to 200,000 µS/cm.	0 to 350,000 μS/cm
Range: Water Level	0 to 5 m	9.14 m			0 to 30 m		0 to 9 m up to 0–250 m
Expanded Uncertainty: pH	±0.4 pH						
Expanded Uncertainty: Conductivity	±57.20 μS/cm at 1413 μS/cm; ±515.20 μS/cm at 12,880 μS/cm; ±3200.00 μS/cm at 80.000 μS/cm						
Expanded Uncertainty: Water Level	±5.50 cm						
Accuracy: pH	0.002 pH		0.15 pH	0.1 pH	0.02 pH	0.02 pH	0.1 pH
Accuracy: Conductivity	2.0%	5% of reading, in waters within a range of 3000 μS cm^−1^, waters with greater variation can have greater error.		5.0%	0.01% of reading	1% of reading or 1 µS/cm, whichever is greater	0.5% of reading plus 1 μS/cm from 0 to 100,000 μS/cm; 1.0% of reading from 100,000 to 200,000 μS/cm; 2.0% of reading from 200,000 to 350,000μS/cm
Accuracy: Water Level	0.5%	0.003			0.16% full scale, 1.5% of reading at constant temp (±2.5 °C); 0.20% full scale, 1.75% of reading from 0 to 30 °C; 0.25% full scale, 2.1% of reading from 0 to 70 °C		0.1% FS from −5 to 50 °C
Power Source	Battery/External (DC)	Battery	Battery	External DC supply	External AC supply	Battery	Battery
Solar-Powered	✓	✗	✓	✗	✗	✗	✗
GSM/GPRS	✓	✗	✗	✗	✓	✗	✓

## Data Availability

Hardware design files (KiCad and Gerbers), firmware (Arduino/C++), server-side code (PHP API + SQL schema), and the web dashboard are available at AQUADAPT_Multiparametric_Probe (GitHub): https://github.com/samuel-q-fernandes/AQUADAPT_Multiparametric_Probe (accessed on 4 November 2025).

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
