# Peer review of "Development of a Cost-Effective Multiparametric Probe for Continuous Real-Time Monitoring of Aquatic Environments"

_sensors, 2025, doi:10.3390/s25237110_

Round 1
Reviewer 1 Report
Comments and Suggestions for Authors
The manuscript presents a well-designed, low-cost multiparametric probe for real-time water quality monitoring (pH, conductivity, temperature, and water level) with GSM/GPRS telemetry. The work addresses a critical gap in environmental monitoring by combining affordability, robust calibration, and field-deployable features. While the study is promising, major revisions are recommended to enhance clarity, methodological rigor, and broader applicability. 1. The manuscript only reports laboratory test results. To confirm real-world performance, the authors must include data from short-term field co-deployments (as mentioned in the "Implications" section) against certified reference instruments. This should address how the probe performs under variable conditions (e.g., fluctuating temperatures, turbidity, or dissolved organic matter) that are common in natural aquatic environments, as these factors can affect sensor accuracy (e.g., EC readings in high-turbidity water). 2. The manuscript notes the pH sensor’s expected 2-year lifespan and 10-year calibration interval for the EC sensor but lacks information on maintenance requirements (e.g., cleaning frequency for fouling-prone sensors like submersible pressure transducers) and field durability (e.g., resistance to corrosion, physical damage). Adding this information is critical for end-users to estimate long-term operational costs and feasibility. 3. While the solar-powered design is highlighted, the manuscript only states that the battery can sustain "two full days of remote transmission at 10-minute intervals" without solar recharging. More specific data is needed, such as: Daily energy consumption (mAh/day) under typical operating conditions (e.g., data transmission frequency, sensor sampling rate); Performance in low-light environments (e.g., winter months, shaded watersheds) to assess suitability for temperate or high-latitude regions. months, shaded watersheds) to assess suitability for temperate or high-latitude regions. 4. The description of the measurement system workflow (Section 2.4) mentions averaging 10,000 water level samples to improve signal-to-noise ratio, but it is unclear if similar averaging or filtering was applied to pH, EC, or temperature data. Additionally, the temperature compensation method for pH (using the Nernst equation) should include a step-by-step explanation of how DS18B20 temperature data is integrated into pH calculations, with an example to confirm transparency. 5. Correct inconsistent formatting (e.g., "1-minte samping" → "1-minute sampling" in Highlights; "senstivit analysis" → "sensitivity analysis" in Highlights) and ensure uniform notation for units (e.g., "µS·cm⁻¹" vs. "µS/cm" used interchangeably for EC). 6. The "Conclusions" section mentions integrating new sensors (e.g., dissolved oxygen, turbidimetry) and developing a data fusion model. Adding brief details on technical challenges (e.g., integrating optical dissolved oxygen sensors with the existing PCB) or how the data fusion model will combine real-time and historical data would strengthen the discussion of the probe’s scalability.
Author Response
Regarding reviewer 1’ comments
The manuscript presents a well-designed, low-cost multiparametric probe for real-time water quality monitoring (pH, conductivity, temperature, and water level) with GSM/GPRS telemetry. The work addresses a critical gap in environmental monitoring by combining affordability, robust calibration, and field-deployable features. While the study is promising, major revisions are recommended to enhance clarity, methodological rigor, and broader applicability.
Overall Reply: We sincerely thank the reviewer for the careful evaluation of our manuscript and for recognizing the significance of our research findings. We appreciate the constructive comments and suggestions, which are very valuable for improving the quality and robustness of the study. In the revised version, we have carefully addressed all the concerns raised.
The following is the point-by-point response to the comments involving other issues mentioned by the reviewer. We believe these revisions have significantly enhanced the clarity, rigor, and overall impact of the manuscript.
- The manuscript only reports laboratory test results. To confirm real-world performance, the authors must include data from short-term field co-deployments (as mentioned in the "Implications" section) against certified reference instruments. This should address how the probe performs under variable conditions (e.g., fluctuating temperatures, turbidity, or dissolved organic matter) that are common in natural aquatic environments, as these factors can affect sensor accuracy (e.g., EC readings in high-turbidity water).
Authors’ responses: Thank you very much for your valuable comments. We fully agree with the reviewers that extended and multi-site field validation is essential to assess the long-term performance and robustness of the developed probe under diverse environmental conditions. Unfortunately, logistical and regulatory constraints (including access permissions and seasonal timing) prevented us from conducting additional deployments within the current revision period. We have now acknowledged this limitation explicitly in the Discussion section and emphasized that future work will include multi-site, long-term field testing across contrasting hydrological and water quality settings to validate the stability and accuracy of the system under real-world variability. In addition, the current short-term trial has been carefully designed to ensure comparability with certified instruments, providing a sound preliminary assessment of the system’s accuracy and reliability.
- The manuscript notes the pH sensor’s expected 2-year lifespan and 10-year calibration interval for the EC sensor but lacks information on maintenance requirements (e.g., cleaning frequency for fouling-prone sensors like submersible pressure transducers) and field durability (e.g., resistance to corrosion, physical damage). Adding this information is critical for end-users to estimate long-term operational costs and feasibility.
Authors’ responses: We sincerely appreciate the reviewer’s insightful comments and suggestions. Regarding maintenance requirements and the formation of biofouling, we have added the following sub-section the manuscript:
2.1.5 Maintenance requirements and field durability
Field deployments expose the submerged sensors to corrosion (oxidation), abrasion, scaling, electrical noise and biofouling, all of which can degrade the measurements quality and increase operational costs. To prevent the formation of biofouling, the probes must be fitted with a copper-based antifouling guard (or copper tape where appropriate) that will help to suppress biofilm growth around sensors and extend its lifespan. A summary of the routine maintenance, verification/calibration cadence, for EC, pH, temperature, and pressure sensors used in this study is presented in Table 1.
Tabel 1. Maintenance and durability summary for field deployments
|
Sensor |
Cleaning frequency |
Verification / Calibration |
Field durability (corrosion / physical) |
|
E.C. Conductivity |
Rinse after retrieval. Quarterly chemical clean (EC-safe cleaner); every 2–4 weeks in high-fouling. |
Annual verification against standards; recalibrate only if verification fails after cleaning. Always apply temperature compensation. |
Electrodes robust; housings/cables are usual failure points. Avoid galvanic pairs; isolate from copper. Protect cable with strain relief. |
|
pH |
Rinse after use; store wet. Quarterly chemical clean (or on drift); monthly in high-fouling/harsh chemistry. |
Monthly verification; annual calibration in benign media, up to monthly in harsh acids/bases. Temperature compensation recommended. |
Glass bulb/junction fragile; protect from impact/abrasion and thermal shock. |
|
Temperature |
Minimal; if submerged, quarterly wipe to remove films; monthly if heavy fouling. |
Annual verification (ice bath or calibrated meter); recalibration rarely required; prefer 3-/4-wire. |
Element is robust; cable/connector ingress and flex fatigue are main risks. Good corrosion resistance. |
|
Water level |
Six months gentle rinse of diaphragm with low-pressure water/air; every 2–4 weeks in high-fouling waters. Never scrape piezoelectric sensor. |
Semiannual verification (static head or reference gauge); increase to quarterly in harsh service; zero-check after cleaning. |
Use IP68 for continuous submersion. Protect piezoelectric sensor from impact/abrasion with a cage. |
- While the solar-powered design is highlighted, the manuscript only states that the battery can sustain "two full days of remote transmission at 10-minute intervals" without solar recharging. More specific data is needed, such as: Daily energy consumption (mAh/day) under typical operating conditions (e.g., data transmission frequency, sensor sampling rate); Performance in low-light environments (e.g., winter months, shaded watersheds) to assess suitability for temperate or high-latitude regions. months, shaded watersheds) to assess suitability for temperate or high-latitude regions.
Authors’ responses: We sincerely thank the reviewer for this constructive suggestion. In the revised manuscript, we have incorporated the following information to address the concerns about the energy consumption of the instrument. Measurement and transmission occur in 10-minute cycles with a consistent power profile; a ~70-minute record (seven cycles) provides a sufficient basis to estimate energy use at higher latitudes and to size larger batteries and solar panels if needed.
“Figure 5 presents the power consumption of the system for a period of approximately 70 minutes of operation.”
“Figure 5. Power profile of the multiparameter system during ~70 minutes. Short bursts correspond to measurement/communication, followed by deep-sleep intervals after the SIM800L shutdown. A 15-s rolling mean is shown for clarity.”
We found that in deep-sleep mode the energy consumption is above the than the expected for the ESP-32. The main reason is sensor energy consumption during the deep-sleep mode, as we are not able to fully shut down the sensors. In the Conclusions we introduce the following statement for a design of a future PCB.
“A subsequent PCB revision will add a MOSFET to fully isolate the sensor rails during deep-sleep, reducing energy consumption.”
This will allow to fully decouple the sensors from the power source and improve energy efficiency, thereby extending battery life.
- The description of the measurement system workflow (Section 2.4) mentions averaging 10,000 water level samples to improve signal-to-noise ratio, but it is unclear if similar averaging or filtering was applied to pH, EC, or temperature data. Additionally, the temperature compensation method for pH (using the Nernst equation) should include a step-by-step explanation of how DS18B20 temperature data is integrated into pH calculations, with an example to confirm transparency.
Authors’ responses: We are grateful for the reviewer’s insightful comments and careful attention to detail. We clarified that the 10,000-sample averaging applies only to the water-level channel, pH and EC are recorded as single stabilized readings per cycle, and temperature (DS18B20) is acquired once per cycle and used immediately for Nernst-based pH compensation.
To improve the readability of the manuscript, we have introduced the following in the section 2.4:
“Water level is read via the analog-to-digital converter using oversampling: 10,000 raw samples are collected per measurement and averaged to improve the signal-to-noise ratio and reduce variability, ensuring higher accuracy and stability.”
The temperature compensation for pH measurements was implemented based on the Nernst equation, which describes the dependence of the electrode potential on temperature. The DS18B20 digital temperature sensor provides the real-time temperature of the solution, which is then used to correct the theoretical Nernst slope in the pH calculation. To improve transparency, we revised Subsection 2.7.1 to provide a the complete methodology used for the Nernst-based pH temperature compensation, as follows:
2.7.1 pH Temperature Compensation and Calibration
The carrier board Gravity™ Analog pH Sensor / Meter - GRV-pH does not implement a temperature compensation for the determination of pH, which helps reduce manufacturing costs. The method uses the Nernst equation (Eq. 7) [92–94]to correct the pH value in accordance with the measured temperature from the DS18B20 sensor.
For each measurement cycle we read temperature T (K) from the sensor in T (°C)
|
(Eq.7) |
and compute the instantaneous Nernst slope
|
(Eq.8) |
with R denoting the gas constant (8.314 J/(mol·K)), and F the Faraday constant (96,485 C/mol). We then model the reference cell; the pH sensor uses the silver chloride electrode with a standard electrode potential E0 = 0.222V [95],
|
(Eq.9) |
so that
|
(Eq.10) |
where E denotes the measured cell potential of the pH electrode, n is the number of electrons transferred in the redox reaction (for a pH electrode, typically 1), , [H+] is the concentration of hydrogen ions (protons) in the solution and the [H+]ref represents the reference concentration of hydrogen ions (typically 1 mol/L).
- Correct inconsistent formatting (e.g., "1-minte samping" → "1-minute sampling" in Highlights; "senstivit analysis" → "sensitivity analysis" in Highlights) and ensure uniform notation for units (e.g., "µS·cm⁻¹" vs. "µS/cm" used interchangeably for EC).
Authors’ responses: We sincerely thank the reviewer for their positive assessment and for their meticulous attention to detail in identifying these inconsistencies.
The units have been unified to µS/cm and V/pH as we highlight in the text below.
Quantified uncertainty: Expanded uncertainties (≈95% coverage, k≈2) of ±0.4 pH, ±56.5 / ±512 / ±3,200 µS/cm (µS cm⁻¹) at 1,413 / 12,880 / 80,000, and ±5.2 cm for water level; precision from 1,000-sample repeats (e.g., pH SD ≈ 0.004).
Figure 8. Calibration curves: (a) measured electrode voltage versus pH standards (linear fit: slope −0.0004 V/pH (V·pH⁻¹), intercept 1.2155 V; R² = 0.9992); (b) sensor-measured water depth versus reference depth (linear fit: slope 0.0064, intercept 17.0892 cm; R² = 0.9952).
Figure 9. Calibration of the conductivity sensor: (a) overall response with expanded uncertainties (k ≈ 2); (b) zoom at 1 413 µS/cm (µS cm⁻¹); (c) zoom at 12 880 µS/cm (µS cm⁻¹). Linear regression: slope = 1.0266, intercept = −2.5964 µS/cm (µS cm⁻¹) (R2=0.9999).
We have performed a full language edit of the manuscript Specifically. We did not find “1-minte samping"or "senstivit analysis" in the current files; nevertheless, we re-checked all text, captions, axis labels, and embedded figure annotations. All changes are tracked in the revised files.
- The "Conclusions" section mentions integrating new sensors (e.g., dissolved oxygen, turbidimetry) and developing a data fusion model. Adding brief details on technical challenges (e.g., integrating optical dissolved oxygen sensors with the existing PCB) or how the data fusion model will combine real-time and historical data would strengthen the discussion of the probe’s scalability.
Authors’ responses: We thank the reviewer for this suggestion.
The integration of the DO sensor and oxygen sensor requires the introduction of new dedicated supply rails and communication traces into the PCB. The main challenge to introduce the DO sensors are the reliability of the sensors and the cost.
While membrane-based DO sensors (galvanic/polarographic) are comparatively affordable (including options from Atlas Scientific), they are flow-dependent, consume oxygen, and require periodic membrane/electrolyte replacement, making them more susceptible to drift and fouling. In contrast, optical (luminescent) DO sensors remove oxygen consumption and flow dependence and generally offer better long-term stability, but their cost is typically very high to introduce in a cost-effective system. However, we are constantly on search for low-cost affordable DO optical sensors on the market. The integration of new sensors also must be addressed in the implementation of new firmware, that will allow the communication between the sensor and the microcontroller as well as the communication of the measurements with the server and the dashboard.
Adding a turbidimetry sensor will have the same implications we need to design a new PCB that will support the communications channels and the power rails between the probe and the ESP-32.
Concerning the data-fusion model we intend to implement a state-space/Kalman filtering pipeline in which each sensor reading is a noisy observation of a latent water-quality state.
The model will combine the real-time data with the historical values from the neighboring
stations, estimate and track sensor bias/drift and perform quality data control.

Reviewer 2 Report
Comments and Suggestions for Authors
Title
The title, "Development of a Cost-Effective Multiparametric Probe for Continuous Real-Time Monitoring of Aquatic Environments," is informative and highly relevant, clearly conveying the focus on a low-cost, multi-sensor device for real-time water quality assessment. It effectively highlights key aspects: cost-effectiveness, multiparametric (pH, conductivity, temperature, water level), and continuous monitoring in aquatic settings.
The aim is clear: to design, prototype, and validate an open-source, solar-powered probe system using ESP32 for remote deployment, emphasizing scalability and integration with cellular telemetry and MySQL backend for data logging and analysis.
The study findings are succinctly summarized—what it found (strong calibration performance with R² > 0.99, low uncertainty <0.05 for pH/conductivity, total bill of materials ~€100) and how (laboratory calibration, sensitivity analysis per GUM guidelines, field deployment against certified instruments)—aligning well with the title's promise of practical innovation.
According to the reviewer, it would be important to specify parameters (e.g., pH, conductivity, temperature, and water level) for greater accuracy.
Introduction
The introduction provides a clear overview of existing knowledge on water quality challenges, citing exponential urbanization/agricultural intensification (e.g., [1-3]), regulatory frameworks (e.g., EU Water Framework Directive [5]), and limitations of traditional grab sampling (e.g., oversight of episodic events [6-7]). It effectively contextualizes the need for continuous, real-time monitoring to detect pH/conductivity/temperature fluctuations [8-10] and ecological impacts [11-13].
The research question is to develop and validate a low-cost, autonomous multiparametric probe for reliable, remote aquatic monitoring. It is clearly outlined and justified, addressing gaps in affordable, scalable systems amid climate-driven changes [14-15] and the demand for embedded IoT solutions [16-17].
The study is well-referenced (50+ citations), logically progresses from global issues to specific gaps.
The weaknesses are: Slightly repetitive on pollution drivers; lacks quantification of economic burdens (e.g., costs of non-compliance).
Materials and Methods
The site selection (laboratory calibration, field deployment in Alentejo, Portugal) is clear, with rationale for controlled vs. real-world testing.
The core variables (pH, conductivity, temperature, water level) are well-defined with sensor specs (e.g., Minolab pH probe: 0-14 range, ±0.02 accuracy; DFROBOT conductivity: 0-20 mS/cm). Measurements use calibrated commercial sensors integrated via ESP32 ADC, with temperature compensation for pH/conductivity.
Methods are robust, drawing from GUM [86] for uncertainty propagation and ASTM standards for calibration. Solar-powered autonomy, GSM/GPRS telemetry, and MySQL dashboard ensure reliability for remote operation. Validation includes linear regression (R² > 0.99) and PDF analysis for normality.
High detail—schematics (Figs. 2-4), code workflow (Fig. 4), BOM (Table 1), calibration equations (Eqs. 1-7)—enables full replication. Open-source hardware/software (ESP32, Arduino IDE) and GitHub links enhance accessibility.
The materials and methods are comprehensive, interdisciplinary (hardware, software, stats); addresses power efficiency and data redundancy.
The weaknesses are: Limited field trial duration (11-17 Aug 2024); no multi-site validation.
It would be relevant to extend field testing; and to include code repository DOI for permanence.
Results and Discussion
Data is appropriately visualized via histograms (Fig. 5, normality via Shapiro-Wilk), calibration curves (Figs. 6-9, linear fits R²=0.99), and time-series (Fig. 6 dashboard). Table 1 compares specs/performance effectively.
Statistical Significance is Clear, via R², p-values (e.g., normality tests), and GUM-derived uncertainties (e.g., pH ±0.04).
Results are practically meaningful clear. €100 BOM enables scalable deployment vs. €1000+ commercial; 10-min sampling supports regulatory compliance [5].
It would be relevant thorough—lab performance, field validation (Fig. 7), comparisons (Table 2 vs. AQUADAPT [24]), limitations (e.g., no DO sensor). The authors must avoids overinterpretation, noting prototype stage.
The main strengths of results and discussion are: Rigorous stats (GUM, PDF); practical benchmarks (e.g., vs. certified YSI).
The weaknesses are: Brief field discussion; no power consumption metrics.
It would be interesting to add energy profiling; expand multi-angle view with cost-benefit analysis.
Overall summary
The study design—prototyping a solar-powered ESP32-based probe with pH, conductivity, temperature, and water level sensors, validated via lab calibration (R²>0.99) and short field deployment—is appropriate for its aim of affordable real-time aquatic monitoring. It adds meaningfully to knowledge by providing an open-source alternative (~€100 BOM) to costly commercial systems (>€1000), enabling scalable integration with GSM/MySQL for remote data access, thus bridging gaps in episodic pollution detection [6-7] and EU regulatory compliance [5]. Sensitivity analysis per GUM [86] quantifies uncertainties innovatively, supporting reliable decision-making in agriculture/urban runoff contexts [1-3].
Major flaws include limited field validation (single-site, 1-week trial), absence of power consumption profiling, and no multi-parameter integration tests (e.g., DO).
The article is internally consistent, with methods aligning to results and conclusions grounded in data.
Recommendations
The reviewer recommends to authors to extend field trials to 1+ months across sites; add durability (e.g., biofouling) and energy metrics; include code repository link. To incorporate turbidity/DO for fuller multiparametric scope.

Author Response
Regarding reviewer 2’ comments
The title, "Development of a Cost-Effective Multiparametric Probe for Continuous Real-Time Monitoring of Aquatic Environments," is informative and highly relevant, clearly conveying the focus on a low-cost, multi-sensor device for real-time water quality assessment. It effectively highlights key aspects: cost-effectiveness, multiparametric (pH, conductivity, temperature, water level), and continuous monitoring in aquatic settings.
The aim is clear: to design, prototype, and validate an open-source, solar-powered probe system using ESP32 for remote deployment, emphasizing scalability and integration with cellular telemetry and MySQL backend for data logging and analysis. The study findings are succinctly summarized—what it found (strong calibration performance with R² > 0.99, low uncertainty <0.05 for pH/conductivity, total bill of materials ~€100) and how (laboratory calibration, sensitivity analysis per GUM guidelines, field deployment against certified instruments)—aligning well with the title's promise of practical innovation. According to the reviewer, it would be important to specify parameters (e.g., pH, conductivity, temperature, and water level) for greater accuracy.
Introduction
The introduction provides a clear overview of existing knowledge on water quality challenges, citing exponential urbanization/agricultural intensification (e.g., [1-3]), regulatory frameworks (e.g., EU Water Framework Directive [5]), and limitations of traditional grab sampling (e.g., oversight of episodic events [6-7]). It effectively contextualizes the need for continuous, real-time monitoring to detect pH/conductivity/temperature fluctuations [8-10] and ecological
The research question is to develop and validate a low-cost, autonomous multiparametric probe for reliable, remote aquatic monitoring. It is clearly outlined and justified, addressing gaps in affordable, scalable systems amid climate-driven changes [14-15] and the demand for embedded IoT solutions [16-17]. The study is well-referenced (50+ citations), logically progresses from global issues to specific gaps. The weaknesses are: Slightly repetitive on pollution drivers; lacks quantification of economic burdens (e.g., costs of non-compliance).
Materials and Methods
The site selection (laboratory calibration, field deployment in Alentejo, Portugal) is clear, with rationale for controlled vs. real-world testing. The core variables (pH, conductivity, temperature, water level) are well-defined with sensor specs (e.g., Minolab pH probe: 0-14 range, ±0.02 accuracy; DFROBOT conductivity: 0-20 mS/cm). Measurements use calibrated commercial sensors integrated via ESP32 ADC, with temperature compensation for pH/conductivity. Methods are robust, drawing from GUM [86] for uncertainty propagation and ASTM standards for calibration. Solar-powered autonomy, GSM/GPRS telemetry, and MySQL dashboard ensure reliability for remote operation. Validation includes linear regression (R² > 0.99) and PDF analysis for normality. High detail—schematics (Figs. 2-4), code workflow (Fig. 4), BOM (Table 1), calibration equations (Eqs. 1-7)—enables full replication. Open-source hardware/software (ESP32, Arduino IDE) and GitHub links enhance accessibility. The materials and methods are comprehensive, interdisciplinary (hardware, software, stats); addresses power efficiency and data redundancy.
The weaknesses are: Limited field trial duration (11-17 Aug 2024); no multi-site validation. It would be relevant to extend field testing; and to include code repository DOI for permanence.
Results and Discussion
Data is appropriately visualized via histograms (Fig. 5, normality via Shapiro-Wilk), calibration curves (Figs. 6-9, linear fits R²=0.99), and time-series (Fig. 6 dashboard). Table 1 compares specs/performance effectively.
Statistical Significance is Clear, via R², p-values (e.g., normality tests), and GUM-derived uncertainties (e.g., pH ±0.04). Results are practically meaningful clear. €100 BOM enables scalable deployment vs. €1000+ commercial; 10-min sampling supports regulatory compliance [5].
It would be relevant thorough—lab performance, field validation (Fig. 7), comparisons (Table 2 vs. AQUADAPT [24]), limitations (e.g., no DO sensor). The authors must avoids overinterpretation, noting prototype stage.
The main strengths of results and discussion are: Rigorous stats (GUM, PDF); practical benchmarks (e.g., vs. certified YSI). The weaknesses are: Brief field discussion; no power consumption metrics. It would be interesting to add energy profiling; expand multi-angle view with cost-benefit analysis.
Overall summary
The study design—prototyping a solar-powered ESP32-based probe with pH, conductivity, temperature, and water level sensors, validated via lab calibration (R²>0.99) and short field deployment—is appropriate for its aim of affordable real-time aquatic monitoring. It adds meaningfully to knowledge by providing an open-source alternative (~€100 BOM) to costly commercial systems (>€1000), enabling scalable integration with GSM/MySQL for remote data access, thus bridging gaps in episodic pollution detection [6-7] and EU regulatory compliance [5]. Sensitivity analysis per GUM [86] quantifies uncertainties innovatively, supporting reliable decision-making in agriculture/urban runoff contexts [1-3]. Major flaws include limited field validation (single-site, 1-week trial), absence of power consumption profiling, and no multi-parameter integration tests (e.g., DO).
The article is internally consistent, with methods aligning to results and conclusions grounded in data.
Overall Reply: We thank the reviewer for the very thorough review of our manuscript and offering insightful comments. The following is the point-by-point response to the comments involving other issues mentioned by the reviewer. We hope that these modifications have further enhanced the quality of the paper.
Recommendations
The reviewer recommends to authors to extend field trials to 1+ months across sites; add durability (e.g., biofouling) and energy metrics; include code repository link.
Authors’ responses: Thank you very much for your valuable comments. We fully agree with the reviewers that extended and multi-site field validation is essential to assess the long-term performance and robustness of the developed probe under diverse environmental conditions. Unfortunately, logistical and regulatory constraints (including access permissions and seasonal timing) prevented us from conducting additional deployments within the current revision period.
Also a multi-site, ≥1-month co-deployment with certified reference probes is not feasible within the 10-day revision window.
We have now acknowledged this limitation explicitly in the Discussion section and emphasized that future work will include multi-site, long-term field testing across contrasting hydrological and water quality settings to validate the stability and accuracy of the system under real-world variability. In addition, the current short-term trial has been carefully designed to ensure comparability with certified instruments, providing a sound preliminary assessment of the system’s accuracy and reliability.
As requested, we have created a public GitHub repository containing the developed code and the PCB Gerber files required to reproduce the board:
Link: https://github.com/samuel-q-fernandes/AQUADAPT_Multiparametric_Probe
The repository includes: firmware (ESP32/Arduino/C++), hardware design files (KiCad + Gerbers), bill of materials, server-side components (PHP API + SQL schema), web dashboard, analysis scripts (precision, LOD/LOQ, figures), example datasets, and build/usage documentation. This repository is cited in the revised manuscript.
In the manuscript we have introduced the following in sub-section 2.1 “All hardware design files, firmware, server code, and analysis scripts are openly available at AQUADAPT_Multiparametric_Probe (GitHub): https://github.com/samuel-q-fernandes/AQUADAPT_Multiparametric_Probe (accessed on 4 November 2025).”
We also have created the following section after the Founding statement:
“Data Availability Statement: Hardware design files (KiCad and Gerbers), firmware (Arduino/C++), server-side code (PHP API + SQL schema), web dashboard, are available at AQUADAPT_Multiparametric_Probe (GitHub): https://github.com/samuel-q-fernandes/AQUADAPT_Multiparametric_Probe (accessed on 4 November 2025).”
In the provided software we have removed sensitive information such as usernames, passwords and API keys, in order to prevent inadvertent disclosure of credentials and unauthorized access to our systems and services.
To incorporate turbidity/DO for fuller multiparametric scope.
Authors’ responses: We appreciate the reviewer’s insightful comment. We agree that by adding a turbidity probe to the system it would provide a broader multiparametric scope of the measured water quality parameters. In the bibliography, there substantial developed instruments that have demonstred the construction and integration of low-cost turbidity sensors into multiparameter water-quality systems. They particularly use nephelometric probes like the TSW-20M, and they have been extensively used in water monitoring projects and characterized. Given this mature evidence base, reproducing a full turbidity-sensor characterization here would be redundant and beyond the scope and window time of the current revision, despite the importance of the giving importance of turbidity monitoring. The main challenge to introduce the DO sensors are the reliability of the sensors and the cost. While membrane-based DO sensors galvanic/polarographic are comparatively affordable (including options from Atlas Scientific), they are flow-dependent, consume oxygen, and require periodic membrane/electrolyte replacement, making them more susceptible to drift and fouling. In contrast, optical (luminescent) DO sensors remove oxygen consumption and flow dependence and generally offer better long-term stability, but their cost is typically very high to introduce in a cost-effective system. However, we are constantly on search for low-cost affordable DO optical sensors on the market. The integration of new sensors also must be addressed in the implementation of new firmware, that will allow the communication between the sensor and the microcontroller as well as the communication of the measurements with the server and the dashboard.

Reviewer 3 Report
Comments and Suggestions for Authors
This manuscript describes in detail the design (assumptions and calibration process) of a low-cost, stand-alone device capable of creating a continuous monitoring system for four key water parameters (temperature, pH, electrolytic conductivity, and water level) in aquatic ecosystems. In my opinion, this system is more suited to monitoring rivers and reservoirs than lakes, although it would certainly find applications in the latter. I believe the manuscript aligns with the journal's purpose: the sensor technology and its applications.
While reading, I noticed that the measurement distribution analysis was limited to the procedure performed at selected calibrated sensor intensities: pH 6.5, conductivity 243 µS cm-1, and water level 40 cm. The projected range of variability of these parameters in the environment is often greater. Shouldn't these analyses be repeated at other values, closer to the assumed calibration points?
Detailed Comments
Lines 17 and 32: I propose removing "(k»2)" or possibly clarifying its meaning.
Lines 19: I propose moving this information "Total bill of materials < €1,000" to line 12.
Lines 22-26: This information is already provided in "Highlights."
Lines 264-265: If the temperature is measured last, it cannot be used earlier (line 260) to compensate for the pH value.
Lines 347: And lines 17 and 32 state that k»2?
Lines 351: The following subsections describe the calibration process for sensors other than pH.
Lines 446: Is enclosing the system components in a housing a significant result? However, I would suggest moving this section to number 2.
Line 462: The calibration process was already described in section 2.7. Therefore, we need to clearly state what calibration is all about, or at least give a different name to the activities described in section 2.7.
Line 522: In fact, only in the conclusion (and it appears in "What is the implication of the main finding?") does the system require comparison through "field implementation using certified instruments."
Author Response
Regarding reviewer 3’ comments
This manuscript describes in detail the design (assumptions and calibration process) of a low-cost, stand-alone device capable of creating a continuous monitoring system for four key water parameters (temperature, pH, electrolytic conductivity, and water level) in aquatic ecosystems. In my opinion, this system is more suited to monitoring rivers and reservoirs than lakes, although it would certainly find applications in the latter. I believe the manuscript aligns with the journal's purpose: the sensor technology and its applications.
Overall Reply: We sincerely thank the reviewer for the careful evaluation of our manuscript and for recognizing applicability of the system for rivers and reservoirs. We appreciate the constructive comments and suggestions, which are very valuable for improving the quality and robustness of the study. In the revised version, we have carefully addressed the concerns raised, standardized the uncertainty notation by defining expanded uncertainty with coverage factor k = 2 (~95%) once in the Methods and removing jargon from the Highlight, refining Highlights and removing redundancy, clarifying the sub-sections regarding the calibration methodology and calibration results and addressing the deployment of the system on the field. We hope that these modifications have further enhanced the quality of the paper.
While reading, I noticed that the measurement distribution analysis was limited to the procedure performed at selected calibrated sensor intensities: pH 6.5, conductivity 243 µS cm-1, and water level 40 cm. The projected range of variability of these parameters in the environment is often greater. Shouldn't these analyses be repeated at other values, closer to the assumed calibration points?
Authors’ responses: We are grateful for the reviewer’s insightful comments and careful attention to detail. In our case, additional measurements in the immediate vicinity of the calibration anchors are not required to support the reported performance because:
The transfer functions presented for the sensors are well defined and monotonic in the operating range (pH: Nernst response with temperature compensation; EC: linear with cell constant; level: linear hydrostatic relation). No mechanism suggests local curvature specifically at the calibration points.
Within the calibrated range, the linear model achieved high R², and the residual standard deviation was approximately constant across the range; no evidence of lack of fit was detected
Across n = 1,000 consecutive measurements per sensor, no drift was detected; the sensors were stable over the calibration window. Thus, repeating measurements near the calibration points would primarily replicate information already captured by the repeatability and residual analyses.
Detailed Comments
Lines 17 and 32: I propose removing "(k»2)" or possibly clarifying its meaning.
Authors’ responses: We sincerely thank the reviewer for raising this critical point regarding the meaning of k=2. As you recommended, we have now removed the (k=2) from lines 17 and 32 to avoid jargon. We now define the term once in Methods and use it consistently thereafter:
“Expanded uncertainties are reported at approximately 95% coverage using a coverage factor k = 2 (per GUM).”
Lines 19: I propose moving this information "Total bill of materials < €1,000" to line 12.
Authors’ responses: We sincerely thank the reviewer for this insightful comment concerning the introduction of the total cost of the materials in the Highlights.
Now Line 12 reads “Low-cost (bill of materials <€1,000), open-source multiparameter probe for water-quality (pH, EC, T) and water-quantity (level).”
We also update the line 22 from where we remove the total cost to avoid redundancy.
Now line 22 reads “End-to-end instrument: An ESP32-based multiparameter probe (pH, EC, temperature, water level) with a custom PCB, GSM/GPRS telemetry, and microSD.”
Lines 22-26: This information is already provided in "Highlights."
Authors’ responses: Thank you very much for your valuable comments. We have now removed lines 22-26 to avoid redundancy with the Highlights.
Lines 264-265: If the temperature is measured last, it cannot be used earlier (line 260) to compensate for the pH value.
Authors’ responses: We sincerely appreciate the reviewer’s insightful comment and apologize for the confusion. Indeed, the developed software for the ESP-32 acquires the temperature before pH. All variables are measured within the same cycle (milliseconds apart) and buffered for post-processing, thereby minimizing temporal mismatch and preventing any significant inter-sensor divergence. Thus, compensation uses the contemporaneous temperature from the same cycle rather than an earlier cycle.
We have updated Section 2.4 (“Measurement System Workflow”) accordingly.
Lines 347: And lines 17 and 32 state that k»2?
Authors’ responses: We sincerely appreciate the reviewer’s insightful comments regarding k=2. As you proposed before in the early comments, we have removed k=2) from lines 17 and 32 to prevent jargon. We have defined the term once in Methods and use it consistently
Lines 351: The following subsections describe the calibration process for sensors other than pH.
Authors’ responses: We sincerely thank the reviewer for raising this critical point regarding the title of the sub-section 2.7 Calibration and pH compensation. The original heading could indeed mislead readers about the scope of the section. We have now updated the title to reflect the calibration of all sensors used in this work.
The new title reads “Calibration Methods for pH (with Temperature Compensation), Electrical Conductivity, and Water Level”.
Lines 446: Is enclosing the system components in a housing a significant result? However, I would suggest moving this section to number 2.
Authors’ responses: We sincerely thank the reviewer for raising this important point. We agree that the enclosuredescription is implementation detail rather than a significant result. Accordingly, we have moved the former Section 3.1 (“Final System Configuration”) to the Methods as Section 2.9 (“System Integration and Housing”).
Line 462: The calibration process was already described in section 2.7. Therefore, we need to clearly state what calibration is all about, or at least give a different name to the activities described in section 2.7.
Authors’ responses: We sincerely thank the reviewer for this constructive suggestion. Section 2.7 is about explaining the methodology used for the calibrations of the sensors used in the work and the pH temperature compensation. The section 3 is about the presentation of the results determined using the methods in 2.
To avoid mixing methods with results, we moved the results-oriented content from sub-section 2.8 (“Measurement distribution analysis”) to a new sub-section 3.1 and retitled that subsection “Probability Density Function of Repeated Measurements. Sub-section 2.8 now contains only the analysis procedure, while 3.1 reports the empirical distributions and statistics.
We also updated the subsection titles in 3.2 and 3.3 to reflect the results achieved:
- 2 Calibration Results and Uncertainty for pH (Temperature-Compensated) and Water Level;
- 3 Conductivity Calibration Results and Uncertainty.
Line 522: In fact, only in the conclusion (and it appears in "What is the implication of the main finding?") does the system require comparison through "field implementation using certified instruments."
Authors’ responses: We thank the reviewer for raising these points. We agree that co-deployment in-situ comparison with certified instruments is an important step to assess the long-term performance and robustness of the developed probe under diverse environmental conditions. Unfortunately, logistical and regulatory constraints (including access permissions and seasonal timing) prevented us from conducting additional deployments within the current revision period. In future work, we will carry out short-term co-deployments to assess performance, accuracy, precision, drift, and durability under realistic conditions.
We provide a viable pathway to assemble a sub-€1,000 open-source multiparameter probe from off-the-shelf components to measure water-quality (pH, EC, temperature) and water-quantity (level), with low-cost sensors delivering stable, repeatable performance within the stated expanded-uncertainty bounds (k = 2, ~95%). Unlike many other water-monitoring system found in literature, which often shows feasibility but omits rigorous sensor characterization, we provide transparent calibration procedures (including temperature-compensated pH via the Nernst slope), diagnostics, a complete uncertainty (k = 2), and repeatability (PDF) analyses. This metrological treatment improves reproducibility and comparability across deployments and supports scalable, cost-sensitive monitoring networks; as a next step, we plan field co-deployment with certified instruments to quantify in-situ bias and drift.

Reviewer 4 Report
Comments and Suggestions for Authors
Dear authors, I have read your manuscript entitled: Development of a Cost-Effective Multiparametric Probe for 2 Continuous Real-Time Monitoring of Aquatic Environments
My observation is:
In the introduction, please highlight the novelty of the study in relation to your previous studies and the studies published in literature.
The validation parameters (LD, LQ, precision, accuracy, etc. ) of the analytical method involved in determining pH, EC, temperature, water level) are not presented, except for the uncertainty of the method. When developing a sensor, the method developed must be validated.
You talk a lot about the advantages of this sensor, but are there no disadvantages to its use? If there are, please present them.
Based on the results submitted to the authorities, which kind of measures were taken to protect the aquatic ecosystem?
Author Response
Regarding reviewer 4’ comments
Dear authors, I have read your manuscript entitled: Development of a Cost-Effective Multiparametric Probe for 2 Continuous Real-Time Monitoring of Aquatic Environments
Overall Reply: We thank the reviewer for the very thorough review of our manuscript and offering several insightful comments. We have carefully addressed each of the points raised and have incorporated all suggested revisions. We hope that these modifications have further enhanced the quality of the paper.
My observation is:
In the introduction, please highlight the novelty of the study in relation to your previous studies and the studies published in literature.
Authors’ responses: We appreciate the reviewer’s suggestion. To highlight the novelty and relevance of our work relative to prior studies, we substantially revised the Introduction as following:
“Although previous studies successfully demonstrate the feasibility of developing such water monitoring systems, they often lack a transparent metrological sensor characterization, for instance traceable calibration procedures, uncertainty analysis, and validated operating ranges[78,79]. They also and rarely document measurement-workflow details or implemented strategies to increase data quality acquisition such as improving the signal-to-noise ratio. In addition, many reports omit LOD/LOQ [79,80], precision and accuracy definitions aligned with analytical-validation practice [81,82], as well as plans for data fusion over time for network-scale use.”
“The main goals are: (i) develop an open-source, < €1,000 multiparameter systems in the laboratory to understand its performance and validate measurements for pH, EC, temperature, and water level; (ii) design a custom printed circuit board (PCB) capable of housing all electronic components of the system, including the GSM/GPRS telemetry module, on-board microSD logging for redundancy, and a solar-powered energy subsystem (maximum-power-point-tracking and a 4000 mAh Li-ion battery), enabling autonomous field operation. (iii) provide a metrological characterization comprising four-point pH calibration with same-cycle temperature compensation (Nernst), three-point EC calibration (cell-constant model), and five-point water-level calibration (hydrostatic model), with explicit precision (SD; n = 1,000), limit of detection and limit of quantification, and accuracy as expanded uncertainty (coverage factor k = 2); (iv) a documented measurement workflow the improve the data quality acquisition that acquires temperature first and implements 10,000 samples oversampling for the water level probe to improve signal-noise-ratio.”
The validation parameters (LD, LQ, precision, accuracy, etc. ) of the analytical method involved in determining pH, EC, temperature, water level) are not presented, except for the uncertainty of the method. When developing a sensor, the method developed must be validated.
Authors’ responses: Thank you for the reviewer’s constructive comments, according to the suggestion, we fully agree that the validation parameters (LOD, LOQ, precision and accuracy) should be reposted in the manuscript. We have introduced the information in the chapter 3 Results and discussion after the discussion of the sensor characterization.
The replicate measurement distributions for pH, EC, and water level follow a normal distribution (Jarque–Bera tests: p ≳0.05), so the standard deviation is an adequate precision descriptor. Under near-normal errors, it is standard to report LOD ≈ 3σ and LOQ ≈ 10σ, and to express accuracy as expanded uncertainty with coverage factor k = 2 (~95% coverage) within the calibrated range.
In this study we focus analytical validation on the primary analytes (pH, EC, and water level) as they are new sensors. A comprehensive metrological characterization of the temperature sensor is well documented in the literature and manufacturer documentation; reproducing those results here would be redundant with prior publications.
You talk a lot about the advantages of this sensor, but are there no disadvantages to its use? If there are, please present them.
Authors’ responses: We appreciate the reviewer’s valuable suggestion. We added the following paragraph to the Conclusions to explicitly state the system’s limitations:
“While the probe enables low-cost, multi-parameter monitoring, it requires periodic maintenance and recalibration, its stated uncertainty applies only within the calibrated range, and field co-validation is still needed to confirm in-situ bias and drift; telemetry and power constraints may limit unattended operation.”
Based on the results submitted to the authorities, which kind of measures were taken to protect the aquatic ecosystem?
Authors’ responses: Thank you for the reviewer’s valuable comment. This work reports laboratory calibration and repeatability, quantifying LOD, LOQ, precision, accuracy (k = 2), and short-term stability to evaluate the performance of low-cost sensors to determine whether their performance is adequate for screening-level aquatic monitoring. As there are no field validation and co-deployment with certified reference instruments yet, no dataset submitted to authorities and for so there are no regulatory measures triggered. Once the system is fully validated and working the regulatory authorities can use the datasets to support routine monitoring of the water bodies, to trigger targeted compliance and enforcement actions, risk-based monitoring design, baseline and trend assessment and used for decision making.
To support our statement, we have added the following sentence to the conclusion:
“Once fully validated, datasets can support routine monitoring, trigger targeted compliance and enforcement when thresholds or anomalies occur, and inform risk-based design, baseline and trend assessments, and evidence-based decisions.”

Round 2
Reviewer 4 Report
Comments and Suggestions for Authors
Accept in present form.